# The Complete Mitochondrial Genome of *Homophyllia bowerbanki* (Scleractinia, Lobophylliidae): The First Sequence for the Genus Homophyllia

**DOI:** 10.3390/genes14030695

**Published:** 2023-03-11

**Authors:** Peng Tian, Wei Wang, Ziqing Xu, Bingbing Cao, Zhiyu Jia, Fucheng Sun, Jiaguang Xiao, Wentao Niu

**Affiliations:** 1Third Institute of Oceanography, Ministry of Natural Resources, Xiamen 361005, China; tianpeng@tio.org.cn (P.T.);; 2Nansha Islands Coral Reef Ecosystem National Observation and Research Station, Guangzhou 510310, China

**Keywords:** evolutionary, phylogenetic, mitogenome, next-generation sequencing

## Abstract

Reef-building coral species of the order Scleractinia play an important role in shallow tropical seas by providing an environmental base for the ecosystem. The molecular data of complete mitochondrial genome have become an important source for evaluating phylogenetic and evolutionary studies of Scleractinia. Here, the complete mitogenome of *Homophyllia bowerbanki* (Milne Edwards and Haime, 1857), collected from Nansha Islands of the South China Sea, was sequenced for the first time through a next-generation sequencing method. *H. bowerbanki* is the first species of its genus for which the mitogenome was sequenced. This mitogenome was 18,154 bp in size and included two transfer RNA genes (tRNAs), 13 protein-coding genes (PCGs), and two ribosomal RNA genes (rRNAs). It showed a similar gene structure and gene order to the other typical scleractinians. All 17 genes were encoded on the H strand and the total GC content was 33.86% in mitogenome. Phylogenetic analysis (maximum likelihood tree method) showed that *H. bowerbanki* belonged to the “Robust” clade and clustered together with other two species in the family Lobophylliidae based on 13 PCGs. The mitogenome can provide significant molecular information to clarify the evolutionary and phylogenetic relationships between stony corals and to facilitate their taxonomic classification; it can also support coral species monitoring and conservation efforts.

## 1. Introduction

Reef-building coral species of the order Scleractinia play a critical role in shallow tropical waters by supporting diverse and abundant marine life and providing an important environmental base for the ecosystem. The molecular data of complete mitochondrial genomes have become an important source for evaluating phylogenetic and evolutionary studies of Scleractinia as the cost of next-generation sequencing (NGS) technology decreases [1,2,3,4]. Multiple-gene analysis of nuclear and mitochondrial genes has already been used to infer phylogenetic relationships amongst scleractinians [5,6]. Nevertheless, only less than 130 complete mitogenomes of Scleractinia species can be obtained through the NCBI database (https://www.ncbi.nlm.nih.gov/ (accessed on 17 February 2023)) to date, even though there are more than 1600 stony coral species [7].

*H. bowerbanki* (Milne Edwards and Haime, 1857) is a species of scleractinian coral with encrusting colonies. Its corallites are cerioid and usually with irregularly angular. Its septa are compact and columellae are small. It usually inhabits lower reef slopes protected from wave action. It shows different colours, such as pale grey, brown, or rust-coloured, and is often mottled [8]. *H. bowerbanki* was placed in the genus *Acanthastrea* by Veron (2000) and Budd et al. (2012) until Arrigoni et al. (2016) transferred it to *Homophyllia.* This genus belongs to the same family as Acanthastrea, Lobophylliidae based on a combination of morphological and three molecular data sets [8,9,10,11].

In this study, the complete mitochondrial genome of *H. bowerbanki* was sequenced and assembled for the first time through a next-generation sequencing method (NGS), in which its genome structure and nucleotide composition were characterized and analysed. Phylogenetic analyses of *H. bowerbanki*, based on 13 protein-coding genes (PCGs) of the mitogenome, will help determine its taxonomic classification and facilitate ongoing taxonomic revisions of scleractinian taxa [12]; analyses can also clarify the evolutionary and phylogenetic relationships amongst stony corals in favor of coral species monitoring and conservation efforts [13].

## 2. Materials and Methods

### 2.1. Sample Collection and DNA Extraction

A specimen (Figure 1) of *H. bowerbanki* was collected from Nansha Islands of the South China Sea (9.9° N, 115.5° E) in 2020. Its fresh polyp was stored at −20 ℃ in 95% ethanol and then used to extract total genomic DNA (gDNA); its skeleton was kept in the Coral Sample Repository of Third Institute of Oceanography, Ministry of Natural Resources, with a unique code, 20200501-L1. The gDNA was extracted using DNeasy Blood and Tissue Kit (Qiagen, Shanghai, China). The integrity and concentration of the gDNA were measured using 1% agarose gel electrophoresis and NanoDrop 2000 spectrophotometer (Thermo Scientific, Waltham, MA, USA).

### 2.2. Mitogenome Sequencing, Annotation, and Analyses

The gDNA sample was sent to Novogene Bioinformatics Technology Co., Ltd. (Beijing, China) for next-generation sequencing. The sequencing library was generated using NEBNext^®^ UltraTM DNA Library Prep Kit for Illumina (NEB, Ipswich, MA, USA, Catalog #: E7370L) according to the standard protocols. Genomic DNA was sheared into fragments about 400 bp using a Focused-ultrasonicator M200 (Covaris, Woburn, MA, USA). Then, DNA fragments were end-polished, A-tailed, and ligated with the full-length adapter for Illumina sequencing, followed by further PCR amplification. Next, PCR products were purified by AMPure XP system (Beckman Coulter, Beverly, MA, USA). Library quality control was performed using Agilent 5400 system (Agilent, Santa Clara, CA, USA) and qPCR (1.5 nM). Then, the library was pooled based on the effective concentration and targeted data amount (12 Gb of raw data). Next, 5′-end of the library was phosphorylated and cyclized. Subsequently, loop amplification was performed to generate DNA nanoballs. These DNA nanoballs were finally loaded into flow cell with DNBSEQ-T7 for paired-end 150 bp sequencing. FastQC was used to assess the quality and quantity of raw data [14]. Fastp was used to remove reads containing poly-N regions, adapters, and low-quality reads; then, a series of filtered clean reads was obtained [15]. These clean reads were applied to reconstruct the mitochondrial genome via NOVOPlasty 4.3.1 [16] with a parameter of K-mer 33. A total of 66,914 of 79,646,476 raw reads (approximately 0.08%) were de novo assembled to produce the mitogenome with the guidance of seed sequence (COI gene from GenBank: MG792550); the average sequencing coverage was 676×.

The circularised contig of *H. bowerbanki* generated by NOVOPlasty 4.3.1 was then submitted to the MITOS Web Server (http://mitos.bioinf.uni-leipzig.de/index.py (accessed on 10 February 2023)) for preliminary annotation [17]. We also identified and annotated PCG and rRNA genes by alignments of homologous mitogenomes from other scleractinians, especially in the same family, through online BLAST searches (https://blast.ncbi.nlm.nih.gov/Blast.cgi (accessed on 10 February 2023)). Meanwhile, we validated the tRNA genes through ARWEN [18]. Finally, the whole mitogenome structure of *H. bowerbanki* was mapped using the online CGView Server (https://proksee.ca/ (accessed on 10 February 2023)) [19]. We obtained base composition, nucleotide frequencies, and codon usage through MEGA11 [20]. The skewing of the nucleotide composition was measured in terms of AT skews ((A − T)/(A + T)) and GC skews ((G − C)/(G + C)) [21,22].

### 2.3. Phylogenetic Analyses

The phylogenetic positions of *H. bowerbanki* were inferred using thirteen tandem mitogenome PCGs together with another forty-one species of Scleractinia and two species of Corallimorpharia (outgroup), which we downloaded from https://www.ncbi.nlm.nih.gov/genbank/ (accessed on 10 February 2023) (Table 1) [22]. We used MEGA11 to select the best-fitting model based on Akaike Information Criterion (AIC); subsequently, a maximum likelihood (ML) tree was constructed with 500 bootstrap replicates under GTR + G + I model.

## 3. Results and Discussion

### 3.1. Characteristics and Composition of Mitogenome

The complete mitogenome of *H. bowerbanki* was identified as a circular molecule with a similar gene order and gene structure to other typical scleractinians [23,24], which included thirteen PCGs (ND5, ND1, Cyt b, ND2, ND6, ATP6, ND4, CO III, CO II, ND4L, ND3, ATP8, and COI), two tRNA genes (tRNA^Met^ and tRNA^Trp^), and two rRNA genes (12S and 16S). The mitochondrial genome size of *H. bowerbanki* was 18,154 bp. The base composition was 24.75% for A, 13.32% for C, 21.75% for G, and 40.17% for T, which showed a higher AT (66.14%) bias. All of the seventeen genes were encoded on H strand and the total GC content was 33.86% in mitogenome. The GC content was one of the important compositional features of different genome regions. Identifying the driving force that shaped the GC content and deciphering the biological meaning of variations in the GC content will help us understand genome evolution [25]. The mitogenome GC content of most scleractinians ranged from 30% to 40% [22,26,27,28]. The intergenic regions of different genes ranged from −1 to 1506 bp. Moreover, four overlaps were detected. ND6 overlapped ATP6 by 1 bp, ATP6 overlapped ND4 by 1 bp, COII overlapped ND4L by 19 bp, and ND5 overlapped tRNA^Trp^ by 2 bp (Figure 2 and Figure 3; Table 2 and Table 3).

### 3.2. Protein-Coding Genes

The total length of all 13 PCGs was 11,595 bp, with a base composition of 21.61% (A), 13.2% (C), 20.56% (G), and 44.63% (T), which showed that the PCGs preferred base AT. All the PCGs started with ATG, except for COIII, COI, and ND2. COIII started with GTG, COI and ND2 started with ATT. Five PCGs (ND1, Cyt *b*, ND4, COII, and ND5) terminated with TAG; the other eight PCGs (ND2, ND6, COIII, ATP6, ND3, ND4L, ATP8, and COI) stopped with TAA. Like other stony corals, the ND5 gene of *H. bowerbanki* also had an intron insertion with a length of 10,461 bp. ATP8 (198 bp) was the shortest gene and the longest gene was ND5 (1,815 bp) (Table 2 and Table 3). According to the analysis results of AT-skew and GC-skew, we can see both were negative (Table 3, Figure 4); meanwhile, all PCGs showed a stronger nucleotide asymmetry, with a higher AT-skew than GC-skew. Codon use frequency was higher amongst L, F, V, G, and S, accounting for 53.04% of a total of 3865 codons (Figure 5).

### 3.3. Transfer RNAs and Ribosomal RNAs

The 12S ribosomal RNA was 910 bp in size and located between ND4 and COIII; the 16S ribosomal RNA was 1,697 bp in size and located between tRNA^Met^ and ND5. The base composition of two rRNAs was 35.52% A, 12.5% C, 20.02% G, and 31.95% T. The AT content of two rRNAs was 67.47%, and both AT-skew (0.053) and GC-skew (0.231) were positive. The two tRNA encoding genes (tRNA^Met^ and tRNA^Trp^) were 72 bp and 69 bp in size, respectively. The base composition of two tRNAs was 32.39% A, 19.01% C, 22.54% G, and 26.06% T. The AT content of two tRNAs was 58.45%, and both AT-skew (0.108) and GC-skew (0.085) were also positive. Both of the two tRNAs were folded into a typical cloverleaf structure, which included an amino acid accept arm, D loop, anticodon loop, and TψC loop (Table 3, Figure 6).

### 3.4. Phylogenetic Analyses

In this study, the 13 PCGs in mitochondrial genomes of *H. bowerbanki* were tandem and established a phylogenetic relationship of Scleractinia. Numbers above the ML tree branches indicated bootstrap percentages (Figure 7). There were three distinct clades (“Complex,” “Robust,” and “Basal”) of Scleractinia in the tree; the ML topology tree showed that *H. bowerbanki* belonged to the “Robust” clade and clustered together with two other species in the family Lobophylliidae based on 13 PCGs. *H. bowerbanki* and *S. maxima* were once placed in the genus *Acanthastrea,* while a molecular method based on several tandem nuclear and mitochondrial DNA has helped to clarify their taxonomic classification [9,10,29,30]. As only fewer than a tenth of mitogenomes of scleractinians can be obtained through NCBI to date, more mitogenomes of other Scleractinia species should be sequenced. Furthermore, the continuous reduction of costs of NGS technologies can facilitate further studies on stony coral evolutionary and phylogenetic relationships and taxonomic classification.

## 4. Conclusions

*H. bowerbanki* is the first species of its genus for which the mitogenome was sequenced. This mitogenome was 18,154 bp in size and included two transfer RNA genes (tRNAs), 13 protein-coding genes (PCGs), and two ribosomal RNA genes (rRNAs). It showed a similar gene structure and gene order to other typical scleractinians. All 17 genes in the mitogenome were encoded on the H strand and the total GC content of the mitogenome was 33.86%, which showed a higher AT bias. Phylogenetic analysis showed that *H. bowerbanki* belonged to the “Robust” clade and clustered together with two other species in the family Lobophylliidae based on 13 PCGs. The mitogenome can provide significant molecular information to clarify the evolutionary and phylogenetic relationships among stony corals and to facilitate their taxonomic classification; it can also support coral species monitoring and conservation efforts.

## Figures and Tables

**Figure 1 genes-14-00695-f001:**
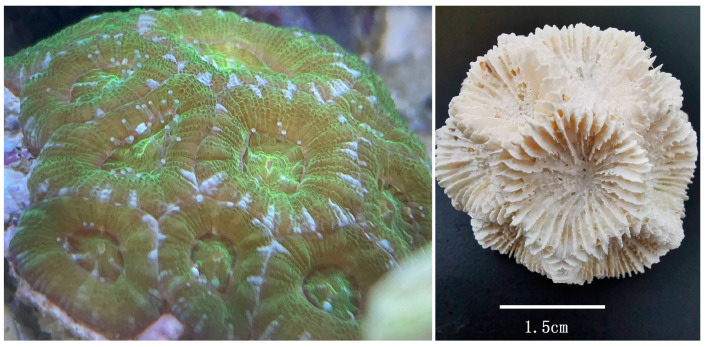
In situ live and skeleton photographs of *H. bowerbanki*.

**Figure 2 genes-14-00695-f002:**
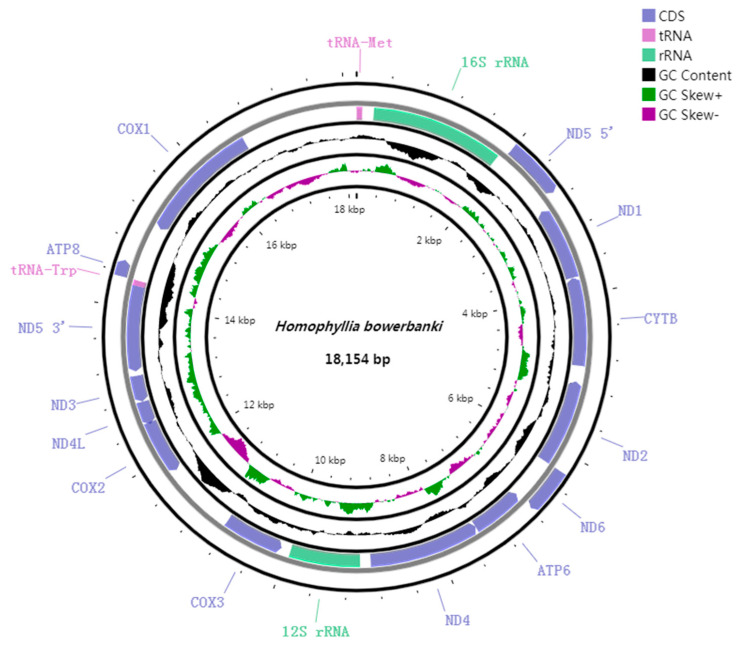
Circular map of the mitogenome of *H. bowerbanki*. COX1, COX2, and COX3 refer to the cytochrome oxidase subunits, ND1–ND6 refer to NADH dehydrogenase components, and CYTB refers to cytochrome b. All of the seventeen genes are encoded on H strand.

**Figure 3 genes-14-00695-f003:**
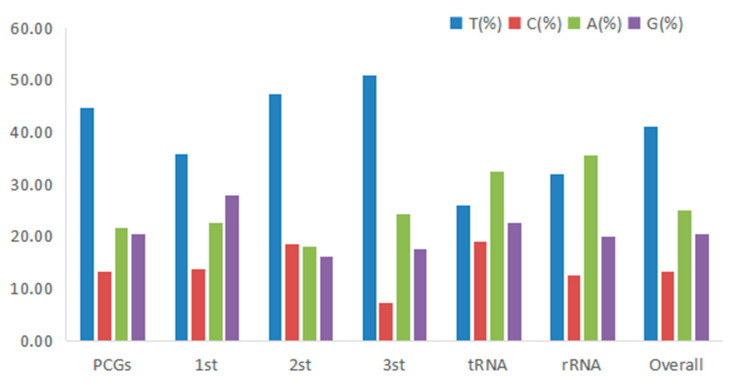
Codon usage bias in different regions of the mitogenome of *H. bowerbanki*.

**Figure 4 genes-14-00695-f004:**
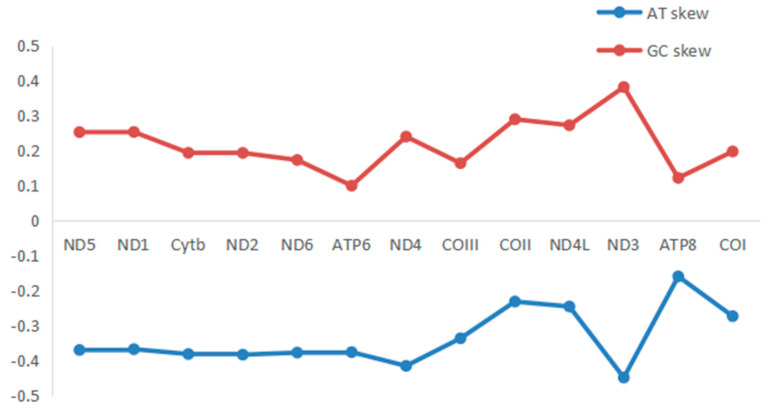
AT−skew and GC−skew of PCGs in mitogenome of *H. bowerbanki*.

**Figure 5 genes-14-00695-f005:**
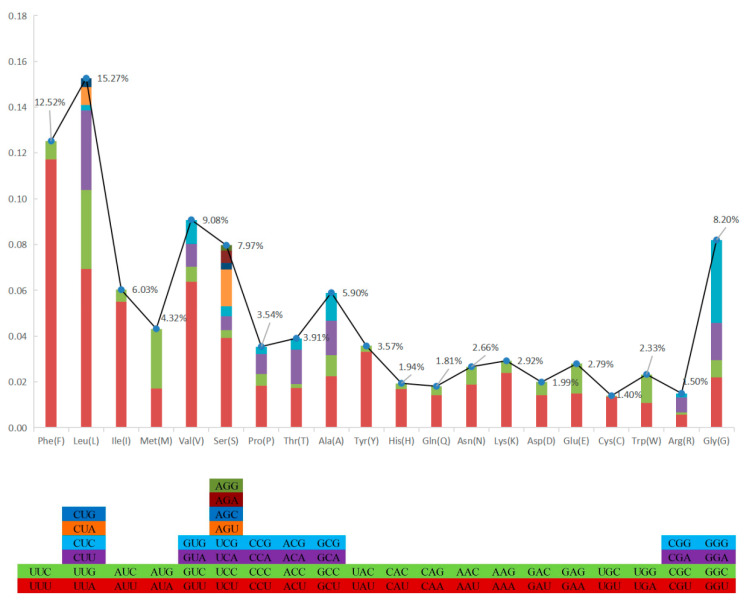
Codon use frequency of PCGs in mitogenome of *H. bowerbanki*.

**Figure 6 genes-14-00695-f006:**
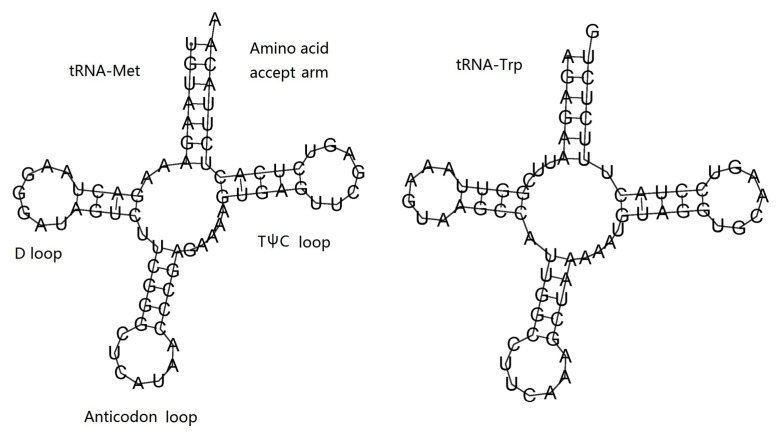
Putative secondary structures of two tRNAs of *H. bowerbanki*.

**Figure 7 genes-14-00695-f007:**
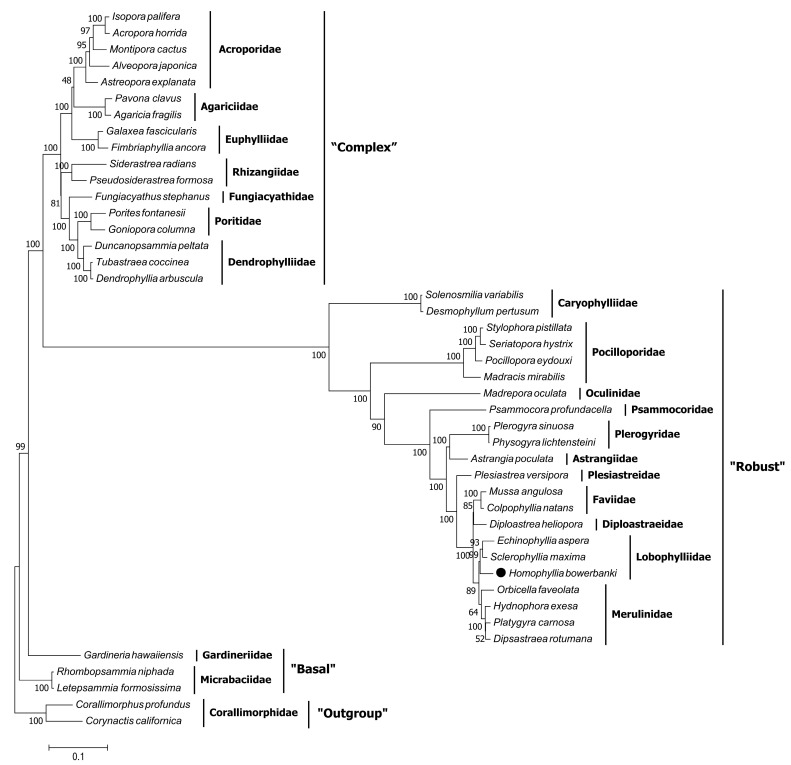
Inferred maximum likelihood phylogenetic tree based on 13 tandem PCGs in mitochondrial genomes of *H. bowerbanki*. Numbers on branches are bootstrap percentages.

**Table 1 genes-14-00695-t001:** Representative Scleractinians and the outgroup species of Corallimorpharia used in the maximum likelihood phylogenetic tree.

No.	Species	Family	Order	Length (bp)	GenBank Accession Number
1	*H. bowerbanki*	Lobophylliidae	Scleractinia	18,154	OP821698
2	*Echinophyllia aspera*	Lobophylliidae	Scleractinia	17,697	MG792550
3	*Sclerophyllia maxima*	Lobophylliidae	Scleractinia	18,168	FO904931
4	*Isopora palifera*	Acroporidae	Scleractinia	18,725	KJ634270
5	*Acropora horrida*	Acroporidae	Scleractinia	18,480	NC_022825
6	*Montipora cactus*	Acroporidae	Scleractinia	17,887	NC_006902
7	*Astreopora explanata*	Acroporidae	Scleractinia	18,106	KJ634269
8	*Alveopora japonica*	Acroporidae	Scleractinia	18,144	MG851913
9	*Fungiacyathus stephanus*	Fungiacyathidae	Scleractinia	19,381	JF825138
10	*Pavona clavus*	Agariciidae	Scleractinia	18,315	NC_008165
11	*Agaricia fragilis*	Agariciidae	Scleractinia	18,667	KM051016
12	*Galaxea fascicularis*	Euphylliidae	Scleractinia	18,751	NC_029696
13	*Fimbriaphyllia ancora*	Euphylliidae	Scleractinia	18,875	NC_015641
14	*Siderastrea radians*	Rhizangiidae	Scleractinia	19,387	NC_008167
15	*Pseudosiderastrea formosa*	Rhizangiidae	Scleractinia	19,475	KP260633
16	*Porites fontanesii*	Poritidae	Scleractinia	18,658	NC_037434
17	*Goniopora columna*	Poritidae	Scleractinia	18,766	JF825141
18	*Tubastraea coccinea*	Dendrophylliidae	Scleractinia	19,094	KX024566
19	*Dendrophyllia arbuscula*	Dendrophylliidae	Scleractinia	19,069	KR824937
20	*Duncanopsammia peltata*	Dendrophylliidae	Scleractinia	18,966	NC_024671
21	*Solenosmilia variabilis*	Caryophylliidae	Scleractinia	15,968	KM609293
22	*Desmophyllum pertusum*	Caryophylliidae	Scleractinia	16,149	KC875348
23	*Seriatopora hystrix*	Pocilloporidae	Scleractinia	17,059	EF633600.2
24	*Stylophora pistillata*	Pocilloporidae	Scleractinia	17,177	NC_011162
25	*Pocillopora eydouxi*	Pocilloporidae	Scleractinia	17,422	EF526303
26	*Madracis mirabilis*	Pocilloporidae	Scleractinia	16,951	NC_011160
27	*Madrepora oculata*	Oculinidae	Scleractinia	15,841	JX236041
28	*Psammocora profundacella*	Psammocoridae	Scleractinia	16,274	MT576637
29	*Plerogyra sinuosa*	Plerogyridae	Scleractinia	17,586	MW936598
30	*Physogyra lichtensteini*	Plerogyridae	Scleractinia	17,286	MW970409
31	*Astrangia poculata*	Astrangiidae	Scleractinia	14,853	NC_008161
32	*Plesiastrea versipora*	Plesiastreidae	Scleractinia	15,320	MH025639
33	*Mussa angulosa*	Faviidae	Scleractinia	17,245	DQ643834
34	*Colpophyllia natans*	Faviidae	Scleractinia	16,906	NC_008162
35	*Diploastrea heliopora*	Diploastraeidae	Scleractinia	18,363	MT560600
36	*Orbicella faveolata*	Merulinidae	Scleractinia	16,138	AP008978
37	*Dipsastraea rotumana*	Merulinidae	Scleractinia	16,466	MH119077
38	*Platygyra carnosa*	Merulinidae	Scleractinia	16,463	JX911333
39	*Hydnophora exesa*	Merulinidae	Scleractinia	17,790	MH086217
40	*Gardineria hawaiiensis*	Gardineriidae	Scleractinia	19,430	MT376619
41	*Rhombopsammia niphada*	Micrabaciidae	Scleractinia	19,542	MT706034
42	*Letepsammia formosissima*	Micrabaciidae	Scleractinia	19,048	MT705247
43	*Corynactis californica*	Corallimorphidae	Corallimorpharia	20,715	NC_027102
44	*Corallimorphus profundus*	Corallimorphidae	Corallimorpharia	20,488	KP938440

**Table 2 genes-14-00695-t002:** Organization of the mitogenome of *H. bowerbanki*.

Sequence	Position		Size (bp)	Amino	Gaps	Codon		Strand
	From	To	Nucleotide	Acid		Start	Stop	
tRNA^Met^	1	72	72		149			H
16s rRNA	222	1918	1697		81			H
ND5 5′	2000	2710	711	237	108	ATG		H
ND1	2819	3766	948	316	2	ATG	TAG	H
Cyt *b*	3769	4908	1140	380	208	ATG	TAG	H
ND2	5117	6220	1104	368	1	ATT	TAA	H
ND6	6222	6782	561	187	−1	ATG	TAA	H
ATP6	6782	7459	678	226	−1	ATG	TAA	H
ND4	7459	8898	1440	480	136	ATG	TAG	H
12S rRNA	9035	9944	910		123			H
CO III	10,068	10,847	780	260	934	GTG	TAA	H
CO II	11,782	12,489	708	236	−19	ATG	TAG	H
ND4L	12,471	12,770	300	100	2	ATG	TAA	H
ND3	12,773	13,114	342	114	57	ATG	TAA	H
ND5 3′	13,172	14,275	1104	368	−2		TAG	H
tRNA^Trp^	14,274	14,343	70		3			H
ATP8	14,347	14,544	198	66	523	ATG	TAA	H
COI	15,068	16,648	1581	527	1506	ATT	TAA	H

Notes: The gaps are the number of nucleotides between the given gene and the related gene behind; negative numbers indicate overlapping nucleotides; H indicates that the genes are transcribed on the heavy strand.

**Table 3 genes-14-00695-t003:** Nucleotide composition in different regions of mitochondrial genome of *H.bowerbanki*.

Gene/Region	T(%)	C(%)	A(%)	G(%)	A + T(%)	Size (bp)	AT Skew	GC Skew
**ND5**	45.84	12.12	21.10	20.33	66.94	1815	−0.370	0.253
**ND1**	44.20	13.19	20.46	22.15	64.66	948	−0.367	0.254
**Cyt b**	45.61	13.68	20.44	20.26	66.05	1140	−0.381	0.194
**ND2**	46.47	13.22	20.74	19.57	67.21	1104	−0.383	0.194
**ND6**	46.88	13.19	21.21	18.72	68.09	561	−0.377	0.173
**ATP6**	46.46	14.60	21.09	17.85	67.55	678	−0.376	0.100
**ND4**	46.18	13.19	19.10	21.53	65.28	1440	−0.415	0.240
**COIII**	41.79	15.64	20.77	21.79	62.56	780	−0.336	0.164
**COII**	40.25	12.29	25.14	22.32	65.39	708	−0.231	0.290
**ND4L**	44.00	10.67	26.67	18.67	70.67	300	−0.245	0.273
**ND3**	49.12	9.94	18.71	22.22	67.83	342	−0.448	0.382
**ATP8**	45.96	9.09	33.33	11.62	79.29	198	−0.159	0.122
**COI**	40.86	14.36	23.34	21.44	64.20	1581	−0.273	0.198
**PCGs**	44.63	13.20	21.61	20.56	66.24	11,595	−0.348	0.218
**1st**	35.83	13.69	22.51	27.97	58.34	3865	−0.228	0.343
**2nd**	47.22	18.65	18.03	16.09	65.25	3865	−0.447	−0.074
**3rd**	50.84	7.24	24.29	17.62	75.14	3865	−0.353	0.417
**tRNA**	26.06	19.01	32.39	22.54	58.45	142	0.108	0.085
**rRNA**	31.95	12.50	35.52	20.02	67.47	2607	0.053	0.231
**Overall**	41.13	13.30	25.01	20.56	66.14	18,154	−0.244	0.214

## Data Availability

The mitochondrial genome of *H. bowerbanki* is available from GeneBank under the accession number OP821698. The associated BioProject, SRA, and Bio-Sample numbers are PRJNA918871, SRR22996606, and SAMN32610146, respectively.

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
