# Peer review of "The Complete Mitochondrial Genome of Homophyllia bowerbanki (Scleractinia, Lobophylliidae): The First Sequence for the Genus Homophyllia"

_genes, 2023, doi:10.3390/genes14030695_

Round 1
Reviewer 1 Report
I have no comments on the contents. There are some mistakes in the grammar, style and spelling. Here are some corrections.
Lines 16-17. A sentence should not start with a genus name abbreviation. The grammar of the sentence is not correct. I propose: Homophyllia bowerbanki is the first species of its genus for which the mitogenome was sequenced. This mitogenome was 18,154 bp in size …
Line 22. “under the” should be “in the”
Line 25. “it can also in favor” should be “it can also be in favor”
Line 27. Keywords should not overlap with title words. Replace: mitochondrial genome; Homophyllia bowerbanki
Line 40. Rephrase “coral with encrusting colonies. Its corallites are cerioid and usually with irregular angular”
Line 42. “they shows” should be “they show”
Line 43. Rephrase “colours such as pale grey, brown, or rust-coloured, and they are often mottled [8].”
Lines 43-44. Rephrase “Homophyllia bowerbanki was placed in the genus Acanthastrea”
Lines 45-47. Rephrase “Arrigoni et al. (2016) transfered it to Homophyllia. This genus belongs to the same family as Acanthastrea, Lobophylliidae, based on a combination of morphological and three molecular data sets [8-11]”
Line 48. “genomes” should be “genome”
Line 50. “its genome structure” should be “in which its genome structure”
Line 54 “stony corals and in favor” should be “stony corals in favor”
Line 59. Please be more specific about the locality: which island or which coastal region
Line 102. “as a circular molecules” should be “as a circular molecule”
Line 103. “with same gene order” should be “with the same gene order”
Line 114. “with base composition” should be “with a base composition”
Lines 160, 161- A sentence should not start with a genus name abbreviation.
Line 162. Rephrase “once been placed in the same genera Acanthastrea” should be “were once placed in Acanthastrea”
Line 163 “helped clarify” should be “helped to clarify”
Line 204. “Echinophyllia Aspera” should be “Echinophyllia aspera” (in italic script)
Line 216. “micromussa” should be “Micromussa” (italic script)
Line 216. Homphyllia (italic script)
Line 216 “lobophylliidae” should be “Lobophylliidae”
Line 220. “Reef Coral Family Lobophylliidae (cnidaria:” should be “reef coral family Lobophylliidae (Cnidaria”
Line 222. “Reef Coral Family Mussidae (cnidaria” should be “reef coral family Mussidae (Cnidaria”
Line 225. “sclerophyllia” should be “Sclerophyllia”
Author Response
Dear reviewer,
For there were no comments on the contents. I have accepted all the corrections about the grammar, style and spelling that you pointed out in my manuscript after careful checking, and I also emphasized the sampling location. For details, please see the revised version of the manuscript.
Thanks very much for your guidance!
Best wishes for you,
Peng Tian
Reviewer 2 Report
The manuscript that has been submitted is well-designed and well-written, and I find it to be quite interesting. However, since this study is primarily descriptive in nature, I have only a few general organizational suggestions to offer.
On the other hand, I think that the phylogenetic reconstruction could benefit from some improvement. While the current method is simple, there are more suitable approaches for analyzing large datasets. In particular, I recommend using PhyML (http://www.atgc-montpellier.fr/phyml/) for phylogenetic reconstruction, which incorporates various support evaluation methods. PhyML provides both parametric and non-parametric support mechanisms, making it an excellent option. Given the strong focus on descriptive analysis in this article, I believe that improving the phylogenetic reconstruction would greatly enhance the broader implications of the manuscript beyond a simple description of mtgenome.
Author Response
Dear reviewer,
Thanks very much for your kindly advice, I appreciate it! PhyML is really a fast approach for analyzing large datasets, it only need less then one hour that we obtained a ML tree. However, the evolutionary tree does not exactly fit the current morphological classification of Scleractinia in my research such as it unable to distinguish outgroups although Scleractinia and Corallimorpharia are totally different order. So, I still use MEGA to analyze evolutionary relationships of scleractinians.
Thanks again and we look forward to hearing from you for any other comments.
Best wishes for you,
Peng Tian